# The Bayesian Case Model: A Generative Approach for Case-Based Reasoning and Prototype Classification

**Been Kim, Cynthia Rudin and Julie Shah**
Massachusetts Institute of Technology
Cambridge, Massachusetts 02139
{beenkim, rudin, julie_a_shah}@csail.mit.edu

## Abstract

We present the Bayesian Case Model (BCM), a general framework for Bayesian case-based reasoning (CBR) and prototype classification and clustering. BCM brings the intuitive power of CBR to a Bayesian generative framework. The BCM learns *prototypes*, the "quintessential" observations that best represent clusters in a dataset, by performing joint inference on cluster labels, prototypes and important features. Simultaneously, BCM pursues sparsity by learning *subspaces*, the sets of features that play important roles in the characterization of the prototypes. The prototype and subspace representation provides quantitative benefits in interpretability while preserving classification accuracy. Human subject experiments verify statistically significant improvements to participants' understanding when using explanations produced by BCM, compared to those given by prior art.

## 1 Introduction

People like to look at examples. Through advertising, marketers present examples of people we might want to emulate in order to lure us into making a purchase. We might ignore recommendations made by Amazon.com and look instead at an Amazon customer's Listmania to find an example of a customer like us. We might ignore medical guidelines computed from a large number of patients in favor of medical blogs where we can get examples of individual patients' experiences.

Numerous studies have demonstrated that exemplar-based reasoning, involving various forms of matching and prototyping, is fundamental to our most effective strategies for tactical decision-making ([26, 9, 21]). For example, naturalistic studies have shown that skilled decision makers in the fire service use *recognition-primed decision making*, in which new situations are matched to typical cases where certain actions are appropriate and usually successful [21]. To assist humans in leveraging large data sources to make better decisions, we desire that machine learning algorithms provide output in forms that are easily incorporated into the human decision-making process.

Studies of human decision-making and cognition provided the key inspiration for artificial intelligence Case-Based Reasoning (CBR) approaches [2, 28]. CBR relies on the idea that a new situation can be well-represented by the summarized experience of previously solved problems [28]. CBR has been used in important real-world applications [24, 4], but is fundamentally limited, in that it does not learn the underlying complex structure of data in an unsupervised fashion and may not scale to datasets with high-dimensional feature spaces (as discussed in [29]).

In this work, we introduce a new Bayesian model, called the Bayesian Case Model (BCM), for prototype clustering and subspace learning. In this model, the prototype is the exemplar that is most representative of the cluster. The subspace representation is a powerful output of the model because we neither need nor want the best exemplar to be similar to the current situation in all possible ways:

for instance, a moviegoer who likes the same horror films as we do might be useful for identifying good horror films, regardless of their cartoon preferences. We model the underlying data using a mixture model, and infer sets of features that are important within each cluster (i.e., subspace). This type of model can help to bridge the gap between machine learning methods and humans, who use examples as a fundamental part of their decision-making strategies.

We show that BCM produces prediction accuracy comparable to or better than prior art for standard datasets. We also verify through human subject experiments that the prototypes and subspaces present as meaningful feedback for the characterization of important aspects of a dataset. In these experiments, the exemplar-based output of BCM resulted in statistically significant improvements to participants' performance of a task requiring an understanding of clusters within a dataset, as compared to outputs produced by prior art.

## 2 Background and Related Work

People organize and interpret information through exemplar-based reasoning, particularly when they are solving problems ([26, 7, 9, 21]). AI Cased-Based Reasoning approaches are motivated by this insight, and provide example cases along with the machine-learned solution. Studies show that example cases significantly improve user confidence in the resulting solutions, as compared to providing the solution alone or by also displaying a rule that was used to find the solution [11]. However, CBR requires solutions (i.e. labels) for previous cases, and does not learn the underlying structure of the data in an unsupervised fashion. Maintaining transparency in complex situations also remains a challenge [29]. CBR models designed explicitly to produce explanations [1] rely on the backward chaining of the causal relation from a solution, which does not scale as complexity increases. The cognitive load of the user also increases with the complexity of the similarity measure used for comparing cases [14]. Other CBR models for explanations require the model to be manually crafted in advance by experts [25].

Alternatively, the mixture model is a powerful tool for discovering cluster distributions in an unsupervised fashion. However, this approach does not provide intuitive explanations for the learned clusters (as pointed out in [8]). Sparse topic models are designed to improve interpretability by reducing the number of words per topic [32, 13]. However, using the number of features as a proxy for interpretability is problematic, as sparsity is often not a good or complete measure of interpretability [14]. Explanations produced by mixture models are typically presented as distributions over features. Even users with technical expertise in machine learning may have a difficult time interpreting such output, especially when the cluster is distributed over a large number of features [14].

Our approach, the Bayesian Case Model (BCM), simultaneously performs unsupervised clustering and learns both the most representative cases (i.e., prototypes) and important features (i.e., subspaces). BCM preserves the power of CBR in generating interpretable output, where interpretability comes not only from sparsity but from the prototype exemplars.

In our view, there are at least three widely known types of interpretable models: sparse linear classifiers ([30, 8, 31]); discretization methods, such as decision trees and decision lists (e.g., [12, 32, 13, 23, 15]); and prototype- or case-based classifiers (e.g., nearest neighbors [10] or a supervised optimization-based method [5]). (See [14] for a review of interpretable classification.) BCM is intended as the third model type, but uses unsupervised generative mechanisms to explain clusters, rather than supervised approaches [16] or by focusing myopically on neighboring points [3].

## 3 The Bayesian Case Model

Intuitively, BCM generates each observation using the important pieces of related prototypes. The model might generate a movie profile made of the horror movies from a quintessential horror movie watcher, and action movies from a quintessential action moviegoer.

BCM begins with a standard discrete mixture model [18, 6] to represent the underlying structure of the observations. It augments the standard mixture model with *prototypes* and *subspace feature indicators* that characterize the clusters. We show in Section 4.2 that prototypes and subspace feature indicators improve human interpretability as compared to the standard mixture model output. The graphical model for BCM is depicted in Figure 1.

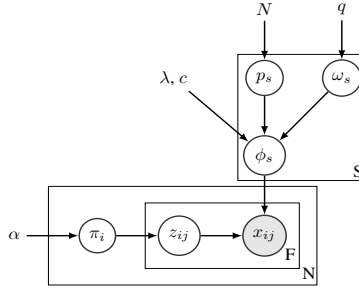

Figure 1: Graphical model for the Bayesian Case Model

We start with $N$ observations, denoted by $\mathbf{x} = \{x_1, x_2, \ldots, x_N\}$, with each $x_i$ represented as a random mixture over clusters. There are $S$ clusters, where $S$ is assumed to be known in advance. (This assumption can easily be relaxed through extension to a non-parametric mixture model.) Vector $\pi_i$ are the mixture weights over these clusters for the $i^{th}$ observation $x_i$, $\pi_i \in \mathbb{R}_+^S$. Each observation has $P$ features, and we denote the $j^{th}$ feature of the $i^{th}$ observation as $x_{ij}$. Each feature $j$ of the observation $x_i$ comes from one of the clusters, the index of the cluster for $x_{ij}$ is denoted by $z_{ij}$ and the full set of cluster assignments for observation-feature pairs is denoted by $\mathbf{z}$. Each $z_{ij}$ takes on the value of a cluster index between 1 and $S$. Hyperparameters $q, \lambda, c$, and $\alpha$ are assumed to be fixed.

The explanatory power of BCM results from how the clusters are characterized. While a standard mixture model assumes that each cluster take the form of a predefined parametric distribution (e.g., normal), BCM characterizes each cluster by a *prototype*, $p_s$, and a *subspace feature indicator*, $\omega_s$. Intuitively, the *subspace feature indicator* selects only a few features that play an important role in identifying the cluster and prototype (hence, BCM clusters are *subspace clusters*). We intuitively define these latent variables below.

*Prototype, $p_s$*: The prototype $p_s$ for cluster $s$ is defined as one observation in $\mathbf{x}$ that maximizes $p(p_s | \omega_s, \mathbf{z}, \mathbf{x})$, with the probability density and $\omega_s$ as defined below. Our notation for element $j$ of $p_s$ is $p_{sj}$. Since $p_s$ is a prototype, it is equal to one of the observations, so $p_{sj} = x_{ij}$ for some $i$. Note that more than one maximum may exist per cluster; in this case, one prototype is arbitrarily chosen. Intuitively, the prototype is the "quintessential" observation that best represents the cluster.

*Subspace feature indicator $\omega_s$*: Intuitively, $\omega_s$ 'turns on' the features that are important for characterizing cluster $s$ and selecting the prototype, $p_s$. Here, $\omega_s \in \{0, 1\}^P$ is an indicator variable that is 1 on the subset of features that maximizes $p(\omega_s | p_s, \mathbf{z}, \mathbf{x})$, with the probability for $\omega_s$ as defined below. Here, $\omega_s$ is a binary vector of size $P$, where each element is an indicator of whether or not feature $j$ belongs to subspace $s$.

The generative process for BCM is as follows: First, we generate the subspace clusters. A subspace cluster can be fully described by three components: 1) a prototype, $p_s$, generated by sampling uniformly over all observations, $1 \ldots N$; 2) a feature indicator vector, $\omega_s$, that indicates important features for that subspace cluster, where each element of the feature indicator ($\omega_{sj}$) is generated according to a Bernoulli distribution with hyperparameter $q$; and 3) the distribution of feature outcomes for each feature, $\phi_s$, for subspace $s$, which we now describe.

*Distribution of feature outcomes $\phi_s$ for cluster $s$*: Here, $\phi_s$ is a data structure wherein each "row" $\phi_{sj}$ is a discrete probability distribution of possible outcomes for feature $j$. Explicitly, $\phi_{sj}$ is a vector of length $V_j$, where $V_j$ is the number of possible outcomes of feature $j$. Let us define $\Theta$ as a vector of the possible outcomes of feature $j$ (e.g., for feature 'color', $\Theta = [\text{red, blue, yellow}]$), where $\Theta_v$ represents a particular outcome for that feature (e.g., $\Theta_v = \text{blue}$). We will generate $\phi_s$ so that it mostly takes outcomes from the prototype $p_s$ for the important dimensions of the cluster. We do this by considering the vector $g$, indexed by possible outcomes $v$, as follows:

$$g_{p_{sj}, \omega_{sj}, \lambda}(v) = \lambda(1 + c\mathbb{1}_{[w_{sj}=1 \text{ and } p_{sj}=\Theta_v]}),$$

where $c$ and $\lambda$ are constant hyperparameters that indicate how much we will copy the prototype in order to generate the observations. The distribution of feature outcomes will be determined by $g$ through $\phi_{sj} \sim \text{Dirichlet}(g_{p_{sj}, \omega_{sj}, \lambda})$. To explain at an intuitive level: First, consider the irrelevant dimensions $j$ in subspace $s$, which have $w_{sj} = 0$. In that case, $\phi_{sj}$ will look like a uniform distribu-

tion over all possible outcomes for features $j$; the feature values for the unimportant dimensions are generated arbitrarily according to the prior. Next, consider relevant dimensions where $w_{sj} = 1$. In this case, $\phi_{sj}$ will generally take on a larger value $\lambda + c$ for the feature value that prototype $p_s$ has on feature $j$, which is called $\Theta_v$. All of the other possible outcomes are taken with lower probability $\lambda$. As a result, we will be more likely to select the outcome $\Theta_v$ that agrees with the prototype $p_s$. In the extreme case where $c$ is very large, we can copy the cluster's prototype directly within the cluster's relevant subspace and assign the rest of the feature values randomly.

An observation is then a mix of different prototypes, wherein we take the most important pieces of each prototype. To do this, mixture weights $\pi_i$ are generated according to a Dirichlet distribution, parameterized by hyperparameter $\alpha$. From there, to select a cluster and obtain the cluster index $z_{ij}$ for each $x_{ij}$, we sample from a multinomial distribution with parameters $\pi_i$. Finally, each feature for an observation, $x_{ij}$, is sampled from the feature distribution of the assigned subspace cluster ($\phi_{z_{ij}}$). (Note that Latent Dirichlet Allocation (LDA) [6] also begins with a standard mixture model, though our feature values exist in a discrete set that is not necessarily binary.) Here is the full model, with hyperparameters $c$, $\lambda$, $q$, and $\alpha$:

$$\omega_{sj} \sim \text{Bernoulli}(q) \;\; \forall s, j \qquad\qquad p_s \sim \text{Uniform}(1, N) \;\; \forall s$$
$$\phi_{sj} \sim \text{Dirichlet}(g_{p_{sj}, \omega_{sj}, \lambda}) \;\; \forall s, j \qquad \text{where } g_{p_{sj}, \omega_{sj}, \lambda}(v) = \lambda(1 + c\mathbb{1}_{[w_{sj}=1 \text{ and } p_{sj}=\Theta_v]})$$
$$\pi_i \sim \text{Dirichlet}(\alpha) \;\; \forall i \qquad\qquad z_{ij} \sim \text{Multinomial}(\pi_i) \;\; \forall i, j \qquad x_{ij} \sim \text{Multinomial}(\phi_{z_{ij}j}) \;\; \forall i, j.$$

Our model can be readily extended to different similarity measures, such as standard kernel methods or domain specific similarity measures, by modifying the function $g$. For example, we can use the least squares loss i.e., for fixed threshold $\epsilon$, $g_{p_{sj}, \omega_{sj}, \lambda}(v) = \lambda(1 + c\mathbb{1}_{[w_{sj}=1 \text{ and } (p_{sj}-\Theta_v)^2 \leq \epsilon]})$; or, more generally, $g_{p_{sj}, \omega_{sj}, \lambda}(v) = \lambda(1 + c\mathbb{1}_{[w_{sj}=1 \text{ and } \ell(p_{sj}, \Theta_v) \leq \epsilon]})$.

In terms of setting hyperparameters, there are natural settings for $\alpha$ (all entries being 1). This means that there are three real-valued parameters to set, which can be done through cross-validation, another layer of hierarchy with more diffuse hyperparameters, or plain intuition. To use BCM for classification, vector $\pi_i$ is used as $S$ features for a classifier, such as SVM.

### 3.1 Motivating example

This section provides an illustrative example for prototypes, subspace feature indicators and subspace clusters, using a dataset composed of a mixture of smiley faces. The feature set for a smiley face is composed of types, shapes and colors of eyes and mouths. For the purpose of this example, assume that the ground truth is that there are three clusters, each of which has two features that are important for defining that cluster. In Table 1, we show the first cluster, with a subspace defined by the color (green) and shape (square) of the face; the rest of the features are not important for defining the cluster. For the second cluster, color (orange) and eye shape define the subspace. We generated 240 smiley faces from BCM's prior with $\alpha = 0.1$ for all entries, and $q = 0.5$, $\lambda = 1$ and $c = 50$.

Table 1: The mixture of smiley faces for LDA and BCM

BCM works differently to Latent Dirichlet Allocation (LDA) [6], which presents its output in a very different form. Table 1 depicts the representation of clusters in both LDA (middle column) and BCM (right column). This dataset is particularly simple, and we chose this comparison because the two most important features that both LDA and BCM learn are identical for each cluster. However, LDA does not learn prototypes, and represents information differently. To convey cluster information using LDA (i.e., to define a topic), we must record several probability distributions – one for each feature. For BCM, we need only to record a prototype (e.g., the green face depicted in the top row, right column of the figure), and state which features were important for that cluster's subspace (e.g., shape and color). For this reason, BCM is more succinct than LDA with regard to what information must be recorded in order to define the clusters. One could define a "special" constrained version of LDA with topics having uniform weights over a subset of features, and with "word" distributions centered around a particular value. This would require a similar amount of memory; however, it loses information, with respect to the fact that BCM carries a full prototype within it for each cluster.

A major benefit of BCM over LDA is that the "words" in each topic (the choice of feature values) are coupled and not assumed to be independent – correlations can be controlled depending on the choice of parameters. The independence assumption of LDA can be very strong, and this may be crippling for its use in many important applications. Given our example of images, one could easily generate an image with eyes and a nose that cannot physically occur on a single person (perhaps overlapping). BCM can also generate this image, but it would be unlikely, as the model would generally prefer to copy the important features from a prototype.

BCM performs joint inference on prototypes, subspace feature indicators and cluster labels for observations. This encourages the inference step to achieve solutions where clusters are better represented by prototypes. We will show that this is beneficial in terms of predictive accuracy in Section 4.1. We will also show through an experiment involving human subjects that BCM's succinct representation is very effective for communicating the characteristics of clusters in Section 4.2.

## 3.2 Inference: collapsed Gibbs sampling

We use collapsed Gibbs sampling to perform inference, as this has been observed to converge quickly, particularly in mixture models [17]. We sample $\omega_{sj}$, $z_{ij}$, and $p_s$, where $\phi$ and $\pi$ are integrated out. Note that we can recover $\phi$ by simply counting the number of feature values assigned to each subspace. Integrating out $\phi$ and $\pi$ results in the following expression for sampling $z_{ij}$:

$$p(z_{ij} = s | z_{i \neg j}, \mathbf{x}, p, \omega, \alpha, \lambda) \propto \frac{\alpha/S + n_{(s,i,\neg j,\cdot)}}{\alpha + n} \times \frac{g(p_{sj}, \omega_{sj}, \lambda) + n_{(s,\cdot,j,x_{ij})}}{\sum_s g(p_{sj}, \omega_{sj}, \lambda) + n_{(s,\cdot,j,\cdot)}}, \quad (1)$$

where $n_{(s,i,j,v)} = \mathbb{1}(z_{ij} = s, x_{ij} = v)$. In other words, if $x_{ij}$ takes feature value $v$ for feature $j$ and is assigned to cluster $s$, then $n_{(s,i,j,v)} = 1$, or 0 otherwise. Notation $n_{(s,\cdot,j,v)}$ is the number of times that the $j^{th}$ feature of an observation takes feature value $v$ and that observation is assigned to subspace cluster $s$ (i.e., $n_{(s,\cdot,j,v)} = \sum_i \mathbb{1}(z_{ij} = s, x_{ij} = v)$). Notation $n_{(s,\cdot,j,\cdot)}$ means sum over $i$ and $v$. We use $n_{(s,i,\neg j,v)}$ to denote a count that does not include the feature $j$. The derivation is similar to the standard collapsed Gibbs sampling for LDA mixture models [17].

Similarly, integrating out $\phi$ results in the following expression for sampling $\omega_{sj}$:

$$p(\omega_{sj} = b | q, p_{sj}, \lambda, \phi, \mathbf{x}, \mathbf{z}, \alpha) \propto \begin{cases} q \times \dfrac{\mathbf{B}(g(p_{sj}, 1, \lambda) + n_{(s,\cdot,j,\cdot)})}{\mathbf{B}(g(p_{sj}, 1, \lambda))} & \text{b} = 1 \\ 1 - q \times \dfrac{\mathbf{B}(g(p_{sj}, 0, \lambda) + n_{(s,\cdot,j,\cdot)})}{\mathbf{B}(g(p_{sj}, 0, \lambda))} & \text{b} = 0, \end{cases} \quad (2)$$

where $\mathbf{B}$ is the Beta function and comes from integrating out $\phi$ variables, which are sampled from Dirichlet distributions.

## 4 Results

In this section, we show that BCM produces prediction accuracy comparable to or better than LDA for standard datasets. We also verify the interpretability of BCM through human subject experiments involving a task that requires an understanding of clusters within a dataset. We show statistically

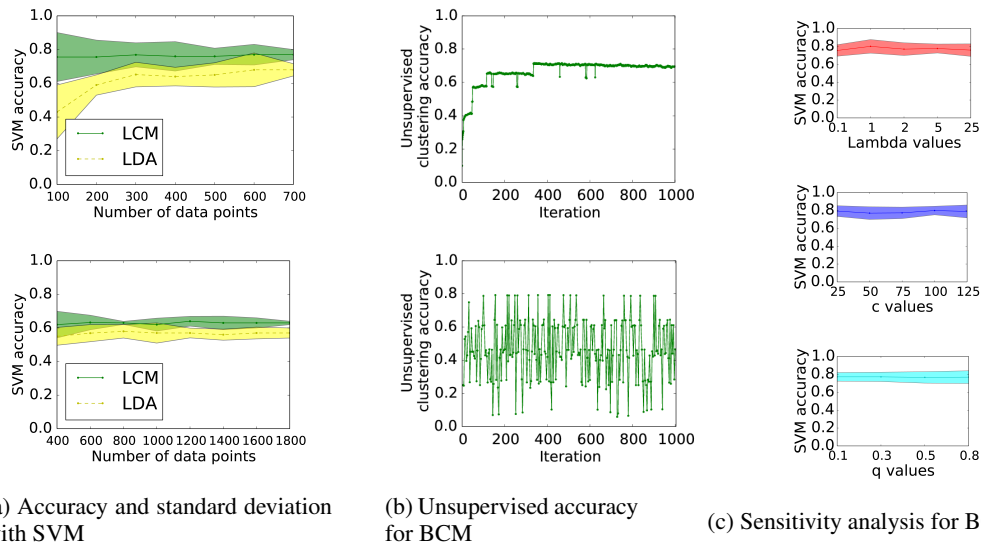

(a) Accuracy and standard deviation with SVM

(b) Unsupervised accuracy for BCM

(c) Sensitivity analysis for BCM

Figure 2: Prediction test accuracy reported for the *Handwritten Digit* [19] and *20 Newsgroups* datasets [22]. (a) applies SVM for both LDA and BCM, (b) presents the unsupervised accuracy of BCM for *Handwritten Digit* (top) and *20 Newsgroups* (bottom) and (c) depicts the sensitivity analysis conducted for hyperparameters for *Handwritten Digit* dataset. Datasets were produced by randomly sampling 10 to 70 observations of each digit for the *Handwritten Digit* dataset, and 100-450 documents per document class for the *20 Newsgroups* dataset. The *Handwritten Digit* pixel values (range from 0 to 255) were rescaled into seven bins (range from 0 to 6). Each 16-by-16 pixel picture was represented as a 1D vector of pixel values, with a length of 256. Both BCM and LDA were randomly initialized with the same seed (one half of the labels were incorrect and randomly mixed), The number of iterations was set at 1,000. $S = 4$ for *20 Newsgroups* and $S = 10$ for *Handwritten Digit*. $\alpha = 0.01, \lambda = 1, c = 50, q = 0.8$.

significant improvements in objective measures of task performance using prototypes produced by BCM, compared to output of LDA. Finally, we visually illustrate that the learned prototypes and subspaces present as meaningful feedback for the characterization of important aspects of the dataset.

## 4.1 BCM maintains prediction accuracy.

We show that BCM output produces prediction accuracy comparable to or better than LDA, which uses the same mixture model (Section 3) to learn the underlying structure but does not learn explanations (i.e., prototypes and subspaces). We validate this through use of two standard datasets: *Handwritten Digit* [19] and *20 Newsgroups* [22]. We use the implementation of LDA available from [27], which incorporates Gibbs sampling, the same inference technique used for BCM.

Figure 2a depicts the ratio of correctly assigned cluster labels for BCM and LDA. In order to compare the prediction accuracy with LDA, the learned cluster labels are provided as features to a support vector machine (SVM) with linear kernel, as is often done in the LDA literature on clustering [6]. The improved accuracy of BCM over LDA, as depicted in the figures, is explained in part by the ability of BCM to capture dependencies among features via prototypes, as described in Section 3. We also note that prediction accuracy when using the full *20 Newsgroups* dataset acquired by LDA (accuracy: $0.68 \pm 0.01$) matches that reported previously for this dataset when using a combined LDA and SVM approach [33]. Also, LDA accuracy for the full *Handwritten Digit* dataset (accuracy: $0.76 \pm 0.017$) is comparable to that produced by BCM using the subsampled dataset (70 samples per digit, accuracy: $0.77 \pm 0.03$).

As indicated by Figure 2b, BCM achieves high unsupervised clustering accuracy as a function of iterations. We can compute this measure for BCM because each cluster is characterized by a prototype – a particular data point with a label in the given datasets. (Note that this is not possible for LDA.) We set $\alpha$ to prefer each $\pi_i$ to be sparse, so only one prototype generates each observation,

Figure 3: Web-interface for the human subject experiment

and we use that prototype's label for the observation. Sensitivity analysis in Figure 2c indicates that the additional parameters introduced to learn prototypes and subspaces (i.e., $q$, $\lambda$ and $c$) are not too sensitive within the range of reasonable choices.

## 4.2 Verifying the interpretability of BCM

We verified the interpretability of BCM by performing human subject experiments that incorporated a task requiring an understanding of clusters within a dataset. This task required each participant to assign 16 recipes, described only by a set of required ingredients (recipe names and instructions were withheld), to one cluster representation out of a set of four to six. (This approach is similar to those used in prior work to measure comprehensibility [20].) We chose a recipe dataset[1] for this task because such a dataset requires clusters to be well-explained in order for subjects to be able to perform classification, but does not require special expertise or training.

Our experiment incorporated a within-subjects design, which allowed for more powerful statistical testing and mitigated the effects of inter-participant variability. To account for possible learning effects, we blocked the BCM and LDA questions and balanced the assignment of participants into the two ordering groups: Half of the subjects were presented with all eight BCM questions first, while the other half first saw the eight LDA questions. Twenty-four participants (10 females, 14 males, average age 27 years) performed the task, answering a total of 384 questions. Subjects were encouraged to answer the questions as quickly and accurately as possible, but were instructed to take a 5-second break every four questions in order to mitigate the potential effects of fatigue.

Cluster representations (i.e., explanations) from LDA were presented as the set of top ingredients for each recipe topic cluster. For BCM we presented the ingredients of the prototype without the name of the recipe and without subspaces. The number of top ingredients shown for LDA was set as the number of ingredients from the corresponding BCM prototype and ran Gibbs sampling for LDA with different initializations until the ground truth clusters were visually identifiable.

Using explanations from BCM, the average classification accuracy was 85.9%, which was statistically significantly higher ($c^2(1, N = 24) = 12.15, p \ll 0.001$) than that of LDA, (71.3%). For both LDA and BCM, each ground truth label was manually coded by two domain experts: the first author and one independent analyst (kappa coefficient: 1). These manually-produced ground truth labels were identical to those that LDA and BCM predicted for each recipe. There was no statistically significant difference between BCM and LDA in the amount of time spent on each question ($t(24) = 0.89, p = 0.37$); the overall average was 32 seconds per question, with 3% more time spent on BCM than on LDA. Subjective evaluation using Likert-style questionnaires produced no statistically significant differences between reported preferences for LDA versus BCM. Interestingly, this suggests that participants did not have insight into their superior performance using output from BCM versus that from LDA.

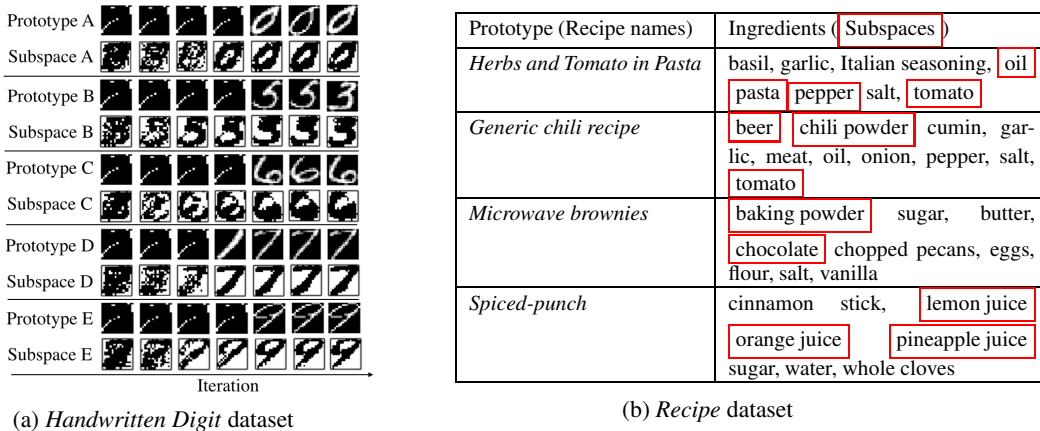

| Prototype (Recipe names) | Ingredients ( Subspaces ) |
|---|---|
| *Herbs and Tomato in Pasta* | basil, garlic, Italian seasoning, oil pasta pepper salt, tomato |
| *Generic chili recipe* | beer chili powder cumin, garlic, meat, oil, onion, pepper, salt, tomato |
| *Microwave brownies* | baking powder sugar, butter, chocolate chopped pecans, eggs, flour, salt, vanilla |
| *Spiced-punch* | cinnamon stick, lemon juice orange juice pineapple juice sugar, water, whole cloves |

(a) *Handwritten Digit* dataset    (b) *Recipe* dataset

Figure 4: Learned prototypes and subspaces for the *Handwritten Digit* and *Recipe* datasets.

Overall, the experiment demonstrated substantial improvement to participants' classification accuracy when using BCM compared with LDA, with no degradation to other objective or subjective measures of task performance.

## 4.3 Learning subspaces

Figure 4a illustrates the learned prototypes and subspaces as a function of sampling iterations for the *Handwritten Digit* dataset. For the later iterations, shown on the right of the figure, the BCM output effectively characterizes the important aspects of the data. In particular, the subspaces learned by BCM are pixels that define the digit for the cluster's prototype.

Interestingly, the subspace highlights the *absence* of writing in certain areas. This makes sense: For example, one can define a '7' by showing the absence of pixels on the left of the image where the loop of a '9' might otherwise appear. The pixels located where there is variability among digits of the same cluster are not part of the defining subspace for the cluster.

Because we initialized randomly, in early iterations, the subspaces tend to identify features common to the observations that were randomly initialized to the cluster. This is because $\omega_s$ assigns higher likelihood to features with the most similar values across observations within a given cluster. For example, most digits 'agree' (i.e., have the same zero pixel value) near the borders; thus, these are the first areas that are refined, as shown in Figure 4a. Over iterations, the third row of Figure 4a shows how BCM learns to separate the digits "3" and "5," which tend to share many pixel values in similar locations. Note that the sparsity of the subspaces can be customized by hyperparameter $q$.

Next, we show results for BCM using the Computer Cooking Contest dataset in Figure 4b. Each prototype consists of a set of ingredients for a recipe, and the subspace is a set of important ingredients that define that cluster, highlighted in red boxes. For instance, BCM found a "chili" cluster defined by the subspace "beer," "chili powder," and "tomato." A recipe called "Generic Chili Recipe" was chosen as the prototype for the cluster. (Note that beer is indeed a typical ingredient in chili recipes.)

## 5 Conclusion

The Bayesian Case Model provides a generative framework for case-based reasoning and prototype-based modeling. Its clusters come with natural explanations; namely, a prototype (a quintessential exemplar for the cluster) and a set of defining features for that cluster. We showed the quantitative advantages in prediction quality and interpretability resulting from the use of BCM. Exemplar-based modeling (nearest-neighbors, case-based reasoning) has historical roots dating back to the beginning of artificial intelligence; this method offers a fresh perspective on this topic, and a new way of thinking about the balance of accuracy and interpretability in predictive modeling.

## Footnotes

[1]Computer Cooking Contest: http://liris.cnrs.fr/ccc/ccc2014/

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
