[Reviews · NeurIPS 2014]

Submitted by Assigned_Reviewer_36

This paper proposes the Latent Case Model (LCM), a Bayesian
approach to clustering in which clusters are represented by a
prototype (a specific sample from the data) and feature
subspaces (a binary subset of the variables signifying those
features that are relevant to the class). The approach is
presented as being a Bayesian, trainable version of the
Case-Based Reasoning approach popular in AI, and is motivated by
the ways such models have proved highly effective in explaining
human decision making. The generative model (Figure 1)
represents each item as coming from a mixture of S clusters,
where each cluster is represented by a prototype and subspace
(as above) and a function \phi which generates features matching
those of the prototype with high probability for features in the
subspace, and uniform features outside it. The model is thus
similar in functionality to LDA but quite different in terms of
its representation. Experimental results are shown for
"prediction accuracy" (more on this later) and unsupervised
accuracy of the clusters vs. a baseline of LDA (Figure 2), and
the results seem superior/comparable to LDA. Further
experiments are shown regarding the interpretability of the
models - human subjects are asked to categorize prototypes from
LCM vs. lists of top features from LDA; the LCM condition
significantly outperformed LDA (85.9% vs. 71.3%).

I greatly enjoyed this paper - not only does it present a novel
method that truly stands out in the vast sea of conventional
LDA-like models, it is refreshingly well-written - the
motivation and background were compelling, the description of
the model and its elements (Section 3) was frankly among the
best I've ever seen, the motivating example (section 3.1) was
clear and extremely helpful, and the experiments were
illustrative (though lacked clarity in places, more details
below). Interpretable models are extremely important, especially
for unsupervised tasks such as clustering - I expect that many
in the NIPS community will find this work compelling and may
want to try this model or variants in their own work.

That said, I do have a few issues, and they have mostly to do
with clarity in the experiments section. First, in Figure 2a,
I'm a bit confused as to what is going on - the authors are
using unsupervised models on digit/newsgroup data, and then
according to the text are training an SVM based on the labels
from those clusters? To what end? And then what is that SVM
being used to classify - some held out data? This is really
unclear, at least to me - the authors do cite [6], which likely
explains it, but really the experimental procedure here should
be fully explained.

The second set of issues have to do with the user study. While
I applaud the authors for having done this study and properly
randomizing orders, conditions, etc., I had a minor concern. On
line 361, when describing the target categories for the task,
the authors say "For LCM, we simply 'presented the prototype'" -
what does that mean - were subjects shown the recipe in its
entirety (all ingredients) or just the *name* of the recipe
(i.e., "fudge brownie")? This needs to be clarified in the
text. Also, if it is the latter (as I suspect, as in Figure 4
the 'Prototype' column is the recipe name), it seems somewhat
unfair to show only a list of top features (ingredients) for the
baseline (LDA) condition - instead they could show a "prototype"
for each cluster there as well by simply selecting the item
that had the highest posterior for that class (this is a common
heuristic to get a representative sample from an LDA topic).
This is not a huge deal, and likely there wouldn't be time to
run the study again, but this should at least be mentioned in
the discussion.

In any case, the issues above are quite minor; I think this is a
significant and well-written paper that would get a good deal of
positive attention at this year's conference.
Summary: This paper presents the Latent Case Model (LCM), a
clustering/topic model whose generative model is based on a
central prototype for each cluster as opposed to a distribution
of words (as in LDA), with the goal of greater interpretability.
The authors show that it performs favorably compared to LDA in
numerical experiments, and then show that it is indeed more
interpretable by humans in a controlled witin-subjects study.
The method is elegant, the paper is beautifully clear, and the
experiments do a good job of supporting the claims (with some
small issues as detailed in the review). Overall, a great paper.

Submitted by Assigned_Reviewer_42

The paper proposes a model that builds upon LDA model, by introducing dependency and coupling of words (observations) in the model. The paper is well written - clear, easy to follow and understand, and shows that proposed model performs better than LDA on several datasets.

Figure 3 is unreadable and needs to be made larger.

Some questions about experiments that should be addressed in the paper:
Why is there poor convergence rate on "newsgroup" dataset? (Figure 2b bottom graph)?
Was the model tried on a very large dataset?

I weakly recommend the paper for acceptance, however ultimately we, as ML community need to ask ourselves: what is the contribution of another variation of a known model and a known parameter estimation algorithm?
Summary: Solid paper introducing a variation of LDA by introducing correlations on words. Weak accept mostly due to originality - the base model is known, and there is nothing new about parameter estimation algorithm.

Submitted by Assigned_Reviewer_45

This paper defines a new generative model, the latent case model (LCM), that forms clusters from prototypes defined on feature subspaces. The proposed benefit of this approach is to combine the merits of case-based reasoning with a Bayesian generative model. The experiments test whether LCM outperforms traditional learning algorithms by preserving classification accuracy and improving human interpretability of the learned clusters. The experimental results provide some support that LCM outperforms LDA along these dimensions.

Quality:
The objective is somewhat confusing and this makes the results less convincing. For a general learning problem, it is unclear why one should be able to make a clustering model that is both accurate and easily interpretable using feature subsets. The paper provides little evidence that combining classification accuracy and human interpretability will scale up to complex problems, and our standard understanding in machine learning argues against it. Historical approaches for developing interpretable models have been largely subsumed by classification with l1-regularizers that provide some tradeoff between interpretability and accuracy through the regularization parameter. Although this approach is motivated through the use of unsupervised learning with clustering and LDA, the primary evaluation in the experiments is through classification. Stronger evidence is needed to advocate for the merits of this approach.

Originality:
This work is novel and unusual in that it attempts to improve the human interpretability of the learned clusters. This is a line of research that has seen little activity in recent years, and this direction of research is worth considering.

Although this work also provides an evaluation of human interpretability for the learned clusters, the description of the experiment does not explain how the prototypes of LCM were better for human interpretation than the top features from LDA. For the examples in Table 1, the subspace features exactly match the top ranked features in LDA (with the prototype having additional features present). In the web interface in Figure 3, we see cluster representatives for one algorithm, but we do not know which algorithm, and we do not know what features the other algorithm provided.

The approach of finding prototypes and explaining the data as a mixture of these clusters has also been explored with non-negative matrix factorizations. It would be interesting to know how the proposed method compares in objectives and performance to the sparse versions of those algorithms, in addition to its performance with LDA.

Clarity:
The paper makes an effort to be easy to read, but the writing style obscured the main idea in places. In the formal description of the generative model, several forward references in definitions and uses of ‘intuitive’ descriptions made the underlying algorithm more difficult to understand.

Significance:
The significance of this paper appears limited. The results show better performance than LDA on some domains, but I do not see a clear argument for the need of human interpretability of underlying clusters in these domains.
Summary: This paper presents a novel approach for integrating some aspects of case-based reasoning within a Bayesian generative framework. However, the arguments for the merits of the proposed method are not very convincing.

Author Feedback
Author rebuttal: Thank you all for your thoughtful comments.

We would like to first point out that our model is not a variant of LDA --- it represents information in a fundamentally different way. The only substantive similarity is that they both start with a mixture model, the rest is entirely different. We agree with Reviewer_36 that our core contribution is a Bayesian, trainable version of the Case-Based Reasoning approach popular in AI. As Reviewer_36 describes - our model is “similar in functionality to LDA” – i.e. it is a clustering algorithm – “but QUITE DIFFERENT in terms of its representation.”

While LDA and our LCM both use a mixture model to represent the underlying distribution of the data, this choice is not the cornerstone of our approach. The power of LCM resides in the representation of the prototype and subspace that are natural explanations of clusters. Table 1 shows this – LCM and LDA represent the same clusters in entirely different ways. LDA provides a weighted distribution over words to define a topic, while LCM provides a prototype and the defining set of words for the cluster. Our work is the first to formalize this concept from case-based reasoning in a trainable, unsupervised learning approach. Our work is also the first to demonstrate through human subject experiments that the prototype and subspace representation provides quantitative, measurable benefits in interpretability. Reviewer 42’s summary of the paper "Solid paper introducing a variation of LDA by introducing correlations on words" is a misstatement of our contributions.

<< On the Human Subject Experiment>>

Reviewer_36 asks whether the name of the prototype recipe (i.e., fudge brownie) and/or the full list of recipe ingredients were shown to each participant in the experiment. For fair basis of comparison with LDA, we do not show the name of the prototype recipe for LCM. As Reviewer_36 correctly points out, doing so would provide an unfair advantage to LCM. In the LCM condition, participants are informed that they are seeing ingredients for an "example recipe", and in the LDA condition they are informed that they are seeing "typical ingredients" of the recipe. If provided the chance, we will clarify this in the final paper submission.

<< Convergence Results in Figure 2b >>

Reviewer_42 asks for clarification on the "poor convergence" in Figure 2b bottom. The fact that consecutive samples vary in accuracy does not indicate "poor convergence." Figure 2b depicts accuracy values that are computed based on samples from the posterior. When converged, samples from posterior are likely to come from their mode(s). If there is only one mode in posterior, most of the samples will come from that mode (as in Fig 2b top). But if there are multiple modes, each sample may come from ANY one of the modes. The expected behavior of the converged Gibbs sampling is to sometimes visit the right mode (i.e., best solution, the highest point in the 2b figure), and sometimes visit other modes (other points in 2b figure). Also note the uniform height of the highest points of the graph: this mode is consistently revisited, which provides support that the Markov chain has indeed converged.

<< Scalability of the algorithm and its interpretability >>

Next we address questions related to scalability of the model in terms of interpretability (Reviewer_45). We agree with Reviewer_45 that the problem of preserving interpretability as complexity of the dataset increases is important. Indeed ell1 regularizers are pervasive currently in the statistics literature, however they handle LINEAR problems only, whereas our model can handle complex nonlinearities. The study of Freitas [1] clearly demonstrates that sparsity is not always a desired property for interpretability. Case-based reasoning has been continually studied since the beginning of AI (see the annual Case-based Reasoning Cooking Contest), and there is an even stronger precedent in AI for reasoning based on cases (e.g., nearest neighbors) for interpretability. We respectfully disagree with Reviewer_45 – although the ell1 methods are linear and scale well to large data sets, they struggle to provide interpretable solutions for these complex problems (e.g. problems solved by SVM or random forests). In this work we apply LCM to a moderately-sized real-world data set with complex nonlinearities (the recipe dataset) and note that LCM statistically significantly improves human subjects’ classification accuracy..

<< Other Comments >>

Reviewer_45’s statement about "significance" states that our results are not significant because the reviewer does not see a clear argument for the need of human interpretability in the domains we studied. The recipe dataset certainly requires human interpretability - humans are the ones searching for (and using) these recipe databases. Millions of people look at recipe websites every day. We also tested on some publicly available standard datasets for the purpose of comparison - those datasets are used in LDA experiments and are quite standard. Our model is significant in that we formalized case-based reasoning into a Bayesian modeling approach. This novel method provides a new strong approach from statistics to solving a traditional AI problem.

Reviewer 36: If provided the chance, we would revise the paper to include the full description of the standard methodology in combining unsupervised clustering and SVM for evaluation [6].
Reviewer 42: We would also improve Figure 3.

[1] Alex A Freitas. Comprehensible classification models: a position paper. ACM SIGKDD Explorations Newsletter, 15(1):1–10, March 2014.